# Nondestructive Detection of Corky Disease in Symptomless ‘Akizuki’ Pears via Raman Spectroscopy

**DOI:** 10.3390/s24196324

**Published:** 2024-09-29

**Authors:** Yue Yang, Weizhi Yang, Hanhan Zhang, Jing Xu, Xiu Jin, Xiaodan Zhang, Zhengfeng Ye, Xiaomei Tang, Lun Liu, Wei Heng, Bing Jia, Li Liu

**Affiliations:** 1School of Horticulture, Anhui Agricultural University, 130 Changjiang West Road, Hefei 230036, China; yangyue1@stu.ahau.edu.cn (Y.Y.); lyrf@stu.ahau.edu.cn (W.Y.); hanzhang@stu.ahau.edu.cn (H.Z.); xujing@stu.ahau.edu.cn (J.X.); yezhenfeng@ahau.edu.cn (Z.Y.); tangxiaomei@ahau.edu.cn (X.T.); liul0921@ahau.edu.cn (L.L.); hengwei@ahau.edu.cn (W.H.); jb1977@ahau.edu.cn (B.J.); 2School of Information and Computer Science, Anhui Agriculture University, 130 Changjiang West Road, Hefei 230036, China; jinxiu123@ahau.edu.cn (X.J.); zxd@ahau.edu.cn (X.Z.)

**Keywords:** ‘Akizuki’ pear, corky disease, nondestructive testing, Raman spectroscopy

## Abstract

‘Akizuki’ pear (*Pyrus pyrifolia* Nakai) corky disease is a physiological disease that strongly affects the fruit quality of ‘Akizuki’ pear and its economic value. In this study, Raman spectroscopy was employed to develop an early diagnosis model by integrating support vector machine (SVM), random forest (RF), gradient boosting decision tree (GBDT), extreme gradient boosting (XGBoost), and convolutional neural network (CNN) modeling techniques. The effects of various pretreatment methods and combinations of methods on modeling results were studied. The relative optimal index formula was utilized to identify the SG and SG+WT as the most effective preprocessing methods. Following the optimal preprocessing method, the performance of the majority of the models was markedly enhanced through the process of model reconditioning, among which XGBoost achieved 80% accuracy under SG+WT pretreatment, and F1 and kappa both performed best. The results show that RF, GBDT, and XGBoost are more sensitive to the pretreatment method, whereas SVM and CNN are more dependent on internal parameter tuning. The results of this study indicate that the early detection of Raman spectroscopy represents a novel approach for the nondestructive identification of asymptomatic ‘Akizuki’ pear corky disease, which is of paramount importance for the realization of large-scale detection across orchards.

## 1. Introduction

‘Akizuki’ pear (*Pyrus pyrifolia* Nakai) is a medium- and late-ripening brown pear that was introduced to Japan [1]. It is known for its juicy and sweet, fine and crisp flesh, unique fragrance, storage resistance, and high quality. Since its introduction, it has been favored by cultivators and consumers and has promising market prospects [2]. However, pear cork disease severely affects the quality of ‘Akizuki’ pear fruit and its commercial economic value, restricting the development of this variety in recent years. Akizuki pear cork disease is a physiological disease that occurs mainly during the fruit expansion period. Several studies have revealed that Ca deficiency is the main cause of pear corky disease [3]. Other mineral elements, such as B, K, and Mg, can also lead to corky disease [4]. Liu reported [5] that the accumulation of lignin in plants leads to the development of corkiness in fruits. For the nutritional level, Cui reported [6] that the cellulose content and starch content in diseased fruits were significantly greater than those in normal fruits, indicating that the cellulose content may be related to the development of corky disease. The disease characteristics of ‘Akizuki’ pear corky disease are divided into two types [7]: the diseased type and the asymptomatic type. The diseased type has obvious brown concave round spots on the surface of the diseased fruit, which the naked eye can observe (Figure 1B,b), whereas the asymptomatic type has no obvious concavity on the surface of the fruit, which may only have subtle dark spots (Figure 1C), while the diseased type is difficult to observe quickly by the naked eye. However, when the fruit is peeled, the flesh has ulcerated brown spots, and the flesh tissue is cork-bound (Figure 1c). At present, the detection of internal diseases is based mainly on manual picking, but this method is laborious, time intensive, labor intensive, and destructive. Thus, enhancing the ‘Akizuki’ pear’s detection and grading technology and ensuring the precision of its quality grading are vital for improving industry competitiveness in the market.

The physical and chemical index detection method is the most accurate and widely used method for fruit detection and grading. However, its operation method is cumbersome and expensive, and it can cause damage to fruit. Therefore, it is more suitable for sampling and not suitable for large-scale fruit detection. Nondestructive testing is the main direction of development for rapid and large-scale detection of fruit. In recent years, research on nondestructive testing technology has focused mainly on machine vision, spectral technology, electronic noses, and ultrasonic measurements [8]. Spectral technology, which includes visible/near-infrared spectroscopy, X-ray, laser Doppler, Raman spectroscopy, and terahertz spectroscopy, is used to detect fruit quality-related information by reflecting, semitransmitting, or fully transmitting through the fruit [9].

Raman spectra are scattering spectra based on the Indian scientist C.V. Raman scattering effect discovered by Raman [10]. When an electromagnetic wave of a certain wavelength is applied to a molecule of the substance under study, it causes a jump in the corresponding energy level of the molecule, producing a molecular absorption spectrum. Raman spectroscopy is an analytical method used to study the scattering of compound molecules by light irradiation, the relationship between the energy level difference between scattered light and incident light, and the vibration frequency and rotation frequency of the compound. This technique provides detailed information on the chemical structure, phase, and morphology, as well as the crystallinity and molecular interactions of the sample. It is widely used in the fields of chemistry, physics, biology, and medicine. Raman spectroscopy has been used primarily in agriculture for testing agricultural product quality [11], detecting crop diseases [12], and identifying pesticide residues [13]. Raman spectroscopy allows for qualitative, structural, and quantitative analyses of fruits and vegetables. For example, it has been used to diagnose Huanglong disease in orange and grapefruit trees [14]; asymptomatic grape Esca disease [15]; rice salt stress; nitrogen, phosphorus, and potassium deficiency [16]; tomato yellow leaf Trichoderma virus (Tylcsv); and tomato spot wilt virus (Tswv) [17]. These experiments provide a theory and rationale for the early detection of ‘Akizuki’ pear corky disease. Raman spectroscopy has several advantages, including nondestructiveness, short detection time, low sample volume, and no need for pretreatment [18].

In recent years, with the rapid development of artificial intelligence technology, machine learning (ML), as one of its core branches, has gradually become an indispensable and effective tool in analytical science. ML has long been used to classify Raman spectral data, using algorithms to extract features from large amounts of data, build models, and then apply those models to make predictions or classifications of new data. Compared with traditional methods, the combination of machine learning algorithms and Raman spectroscopy has significant advantages in terms of data processing and analysis ability, model construction and prediction ability, expansion of application areas, adaptability and flexibility. These advantages make the combination of machine learning algorithms, and Raman spectroscopy have a wide range of application prospects and important scientific value in a variety of fields, such as analytical science, food safety, and biomedicine. In the field of food safety applications, the molecular structure information of food samples is obtained via Raman spectroscopy and combined with machine learning algorithms for rapid identification and classification, which can effectively detect harmful substances or adulterated ingredients in food. For example, Zhang et al. [19] and Yan et al. [20] used Raman spectroscopy in combination with ML to achieve accurate classification and analysis of foodborne pathogens, and Zhao et al. [21] compared ML with the principal component analysis (PCA) model, which is more efficient and accurate for the rapid identification of the type of edible oils and adulteration ML. In the field of medicine, Raman spectroscopy combined with machine learning is used for rapid analysis of drugs and improved identification of various diseases [22]. In summary, the combination of Raman spectroscopy and machine learning has significant advantages and broad prospects for development. In the future, with the continuous progress of technology and the continuous expansion of application fields, this combination will play an important role in more fields and promote the innovation and development of related technologies.

The spectral data are affected by many factors in the acquisition process, such as cosmic rays, the fluorescent background, and the dark current of the detector. These factors may lead to noise and interference in the spectral data, which may affect the accurate extraction of the chemical structure, phase and morphology information of the samples. Therefore, before Raman spectroscopy analysis, spectral data must be preprocessed to eliminate these interfering factors as much as possible to ensure the analytical accuracy and modeling stability of the collected data. While a single data preprocessing method often has difficulty in achieving the desired processing effect due to its limitations, the joint use of preprocessing methods has emerged to combine them and play a complementary role [23]. Therefore, six preprocessing methods and three combinations were chosen to preprocess the original spectral data. Then, their impact on modeling accuracy was examined to determine the most suitable spectral preprocessing method.

This study investigated the feasibility of Raman spectroscopy for the nondestructive detection of corky disease in ‘Akizuki’ pears. A Raman spectrometer was used to collect the Raman spectrum. By the Savitzky-Golay (SG), wavelet transform (WT), first derivative (FD), second derivative (SD), standard normal variable (SNV), multiple scattering correction (MSC), and three combination methods (SG+MSC, SG+SNV, and SG+WT), the original spectrum was preprocessed to remove the interference background. Five modeling methods, namely, support vector (SVM), random forest (RF), gradient lifting decision tree (GBDT), extreme gradient enhancement (XGBoost), and convolutional neural network (CNN), were used to establish a classification model for ‘Akizuki’ pear cork disease, and the most suitable classification model for the disease was selected. Ultimately, the performance of the model was optimized by adjusting the parameters of the most effective preprocessing model.

## 2. Materials and Methods

### 2.1. Material Preparation

In September 2023, a test was conducted at the ‘Akizuki’ pear demonstration base in Peach and Pear World Agricultural Fruit City, Shushan District, Hefei City, Anhui Province. The test samples consisted of 30 healthy fruits, 30 fruits with symptomatic disease, and 30 fruits with asymptomatic disease—all of which were selected under the guidance of experienced local fruit farmers. After being transported to the laboratory at the Biotechnology Building of Anhui Agricultural University, the samples were cleaned, wiped, and numbered for future use.

### 2.2. Spectral Data Acquisition

#### 2.2.1. Instrumentation

The ultraminiature Raman spectroscopy identifier ‘ATR1600’, produced by AuPuTianCheng (Xiamen, Fujian, China) Optoelectronics Co., Ltd., was used to collect Raman spectral data in this experiment. The instrument has a spectral laser wavelength of 785 ± 0.5 nm, a wavelength range of 200–3000 cm^−1^, and a spectral resolution of 18 cm^−1^. To collect the spectral data, the ‘ATR1600′ application on a mobile phone is connected to the instrument via Bluetooth, and the mobile phone app is used, as shown in Figure 2.

#### 2.2.2. Data Collection

Raman spectral data were collected separately for different types of ‘Akizuki’ pear fruits. To reduce interference from other factors, the temperature of the room was maintained at approximately 20 °C, and the humidity was kept constant during spectral collection. First, for diseased fruits, the sampling area was selected directly by the location of the surface spots (Figure 3a), whereas for pear fruits without surface spots, the sampling area was selected at 120° intervals near the equator line of the pear equator with similar sizes (Figure 3b). A five-point sampling method was adopted in each sampling area, and the average value was taken as the scanning result. The pear fruits were then peeled to screen diseased pears without obvious spots on the surface but with brown spots of ulcer disease on the inside as asymptomatic data.

### 2.3. Spectral Data Preprocessing

The laboratory data are susceptible to environmental, sample, and equipment factors and other natural and unnatural factors. This can result in a spectral curve model that contains interference and disordered data, which can affect the accuracy and stability of the curve. Therefore, selecting an appropriate preprocessing method is crucial for ensuring the accuracy and stability of subsequent spectral curve establishment and modeling. Therefore, six preprocessing methods and three combinations, including multiple scattering correction (MSC), standard normal variable (SNV), first-order derivative (FD), second-order derivative (SD), Savitzky-Golay smoothed filtering function (SG), and wavelet transform (WT), SG+SNV, SG+MSC, and SG+WT, were used in the present study.

Scattering correction mainly includes MSC and SNV, which are employed to eliminate the influence caused by irregular particle size and distribution, eliminate interference and deviation caused by the scattering effect, and improve the quality and stability of spectral data [24]. The MSC aims to mathematically remove the spectral variability caused by the physical characteristics of the sample, such as the size of the particles. Unlike the MSC, the SNV corrects each spectrum individually. Instead of using a reference spectrum, SNV calculates the standard deviation for each wavelength and uses it to normalize the raw spectral data. Derivative processing is a common technique used in spectral analysis to extract spectral features and reduce the effects of baseline shifts and overlapping peaks. FD indicates changes in the slope of the spectrum and can reveal peaks and troughs. The SD shows changes in spectral curvature and is useful for precise feature extraction and noise amplification [25]. SG is a smoothing technique that reduces noise in spectral data while preserving signal features. This algorithm fits a polynomial function within a window around the data points, preserving peak heights and widths [26]. The WT is able to effectively remove noise and extract useful signals through its localization analysis property and denoising by the wavelet thresholding method, which has unique advantages in spectral denoising [27]. Among the three preprocessing combinations of the SG+MSC, SG+SNV and SG+WT algorithms, the SG and WT smoothing algorithms can eliminate noise burrs in the spectra, whereas the MSC and SNV algorithms can correct the spectral intensity variations. Together, these processing steps improve the quality of the spectral data and provide a more reliable database for subsequent analysis.

### 2.4. Relative Optimal Preprocessing Method Sorting Method

To select the optimal preprocessing method among the nine preprocessing methods of the ‘Akizuki’ pear corky disease prediction model, this paper refers to the relative optimality index formula in Equation (1) to calculate the difference between different preprocessing methods.
(1)$${ROI\mathrm{ndex}}_{j}=\sum_{i=1}^{n}({RA}_{\mathrm{ij}}+RK_{\mathrm{ij}})$$

In Formula (1), ROIndexj is the relative optimal index, j represents the preprocessing method, and i is the model type. When model i is preprocessed as j, its accuracy is ranked as RAij in the list; when model i is preprocessed as j, its kappa values are ranked as RKij. The ranking is in ascending mode. To ensure the degree of universality of the preprocessing method, the ROIndexj formula adds the ranking of each model in the precision list and the kappa list when the preprocessing is j. This results in a ranking list that takes into account both the model accuracy and the classification consistency level. The relative best pretreatment technique can be chosen with clarity through data comparison [28].

### 2.5. Modeling Approach

#### 2.5.1. Data Sampling

Selecting representative sample data and differentiating data are crucial steps in spectral data processing. These steps reduce the modeling effort and increase model stability and accuracy. The KS method is a commonly used algorithm for data sampling and sample set selection, particularly in spectroscopy and chemical analysis for model construction and calibration. The algorithm’s primary objective is to select representative samples from a dataset for constructing models, calibrating instruments, or performing analyses. This method reduces the dataset size and improves model efficiency while maintaining dataset diversity [29]. Following the KS method, the total number of samples was 345 (Healthy 0: 87; Asymptomatic 1: 90; Diseased 2: 168), train:test = 7:3 (241:104), where train: 0:1:2 = 65:62:114 Test: 0:1:2 = 22:28:54, as shown in Table 1.

#### 2.5.2. Modeling Methods

With the growth and development of artificial intelligence in recent years, machine learning has manifested great advantages, which greatly accelerate the speed of experimental analysis and computation by effectively learning from a large amount of prelabeled data and then generating reasonable predictions for new datasets, making it an effective tool for analyzing Raman spectral data [30]. The support vector (SVM), random forest (RF), extreme gradient boosting (XGBoost), gradient-boosted decision tree (GBDT), and convolutional neural network (CNN) methods are the comparative modeling methods employed in this study.

SVM is a boundary-based supervised learning method that is widely applied to classification and regression analysis. In spectral data analysis, SVM correctly divides the training dataset with the most efficient geometric intervals by finding a hyperplane that can classify the data into different classes by choosing appropriate support vectors [31]. By projecting the samples to a high-dimensional feature space via the kernel, the SVM can create the ideal separation hyperplane when the samples are indistinguishable. The performance of the SVM model is significantly influenced by the selection of the kernel function, kernel parameters, and penalty parameters. Different parameters and kernel functions may lead to completely different results.

Integrated modeling is an important approach within the domain of machine learning and is a machine learning model that integrates the prediction results of multiple weak classifiers or weak regressors by combining them via a strategy to obtain a more powerful and robust prediction performance [32]. RF is a typical application of the bagging algorithm in integrated learning and one of the most popular algorithms in the current field of machine learning. Multiple decision trees are used to train and predict samples, and their predictions (regression problem) or majority voting (classification problem) are averaged to obtain the final prediction [33]. RF excels at handling high-dimensional data, easily performs parallel processing, works well with large sample training, and is robustly detect missing data and nonlinear relationships, but is prone to overfitting in noisy datasets. 

GBDT is an iterative approach for training multiple weak classifiers (decision trees) and then combining them into one strong classifier. In GBDT, each decision tree is a correction for the prediction error of the current model, and in this way, the prediction performance of the model is gradually improved [34]. GBDT performs well in all kinds of machine learning tasks because of its efficient running speed and excellent prediction performance, especially when dealing with regression problems and some complex classification problems. XGBoost is based on GBDT to construct new decision trees in the negative gradient direction of the loss function. The generation process of each decision tree is an optimization problem, and constructing the decision tree by iteratively choosing the best split point is a successful execution of GBDT. In contrast to GBDT, XGBoost controls model complexity and guards against overfitting by adding regularization elements to the loss function. Given that calculating the derivative of the loss function is difficult, XGBoost utilizes not only the first derivative of the loss function but also the second derivative (Hessian matrix) to speed up convergence and improve the accuracy of the model.

Deep learning represents a branch of machine learning research that uses neural networks and relies on training data to increase accuracy. Compared with traditional machine learning methods, deep learning has excellent learning ability and low generalization error. The CNN is an exemplary deep learning algorithm that contains convolutional computations and has the deep structure of a feedforward neural network. The basic layer is the core layer of the CNN, and its principal function is to perform convolutional operations on the input 2D image to extract the characteristics of the picture. Pooling layers are used to reduce the spatial size of the data, minimize the number of parameters and computations in the network, and prevent overfitting. On the other hand, the fully connected layer is located at the end of the CNN and classifies the feature vector outputs from the pooling layer, mapping them to the probabilities of each category. It automatically extracts features from the data by simulating the human brain’s processing of visual information, using structures such as convolutional and pooling layers to achieve tasks such as classification and recognition [35].

### 2.6. Evaluation Metrics

The output of machine learning algorithms needs to be evaluated and analyzed, and this analysis must be interpreted correctly, so evaluation metrics are critical to understanding the performance of the model. For classification problems, binary classification models are generally evaluated via a confusion matrix. The actual values of the confusion matrix of the binary classifier (as shown in Table 2) are labeled “true” (1) and “false” (0), and the predicted values are labeled “positive” (1) and “negative” (0). The expressions TP, TN, FP, and FN in the confusion matrix provide the probability estimate for the classification model, where TP (true positive) refers to the case where the positive category is accurately predicted by the model to be positive; FP (false positive) refers to a situation in which the model incorrectly predicts a negative category to be positive; FN (false negative) refers to a situation in which the model incorrectly predicts a positive category to be negative; and TN (true negative) refers to the situation where the model correctly predicts a negative category as a negative category [36].

The following four metrics provide a comprehensive picture of the performance of the classification model, including accuracy, accuracy, recall, and F1 score.
(2)$$Accuracy=\frac{TP+TN}{TP+TN+FN+FP}$$
(3)$$F1=\frac{2TP}{2TP+FP+FN}$$
(4)$$Kappa=\frac{Po-Pe}{1-Pe}$$
(5)$$Po=\frac{TP+TN}{TP+TN+FN+FP}$$
(6)$$Pe=\frac{\frac{\left(TP+FP)\times (TP+FN\right)}{n}+\frac{\left(FN+TN)\times (TN+FP\right)}{n}}{n}$$

The most intuitive categorical metric is accuracy, which indicates the proportion of total samples correctly predicted by the model. The percentage of samples that the model correctly predicts to fall into the positive category out of all samples that the model predicts to fall into the positive category is known as precision. The precision and recall harmonic average is the F1 score and is a composite representation of them. The kappa coefficient is a measure of statistical consistency used to evaluate the consistency of the results of two or more evaluators on the same set of samples or to evaluate the consistency of the model’s predicted and actual classification results in a classification problem, where P0 represents the total accuracy of the module, and PE represents the random accuracy of the model. The range of the kappa values is −1–1. A score of −1 denotes total disagreement. A value of 0 denotes random agreement, i.e., there is no systematic correlation between the predicted and actual results, as in a random guess. A value of 1 indicates a perfect prediction, i.e., the predicted results are in perfect agreement with the actual results. Table 3 shows the corresponding kappa values for categorical agreement at different intervals [37].

## 3. Results and Discussion

### 3.1. Dataset Statistics

The variation in the peak values of the spectral curve is the result of the combined effects of multiple factors, including the chemical composition, physical state of the sample, and conditions of spectral detection. By analyzing the spectral curve, we can obtain important information about the sample’s composition, structure, and properties. Raman spectroscopy was established by collecting data from the surface of the ‘Akizuki’ pear. The cell wall of the pear epidermis is a thick-walled structure that exists outside of plant cells, and it is closely related to the growth, development, and life functions of the cells. ‘Akizuki’ pear corky disease causes changes in the physicochemical properties of pear epidermal cells, and several papers have shown that pear corky disease leads to a significant increase in the content of lignin and cellulose in the cell wall [5,6]. Lignin and cellulose are important components of the cell wall, and they work together to maintain the structural stability and function of the cell wall. Therefore, the resolution of the Raman characteristic region of cellulose and lignin is crucial for analyzing and verifying the altered physicochemical properties of the corky disease in ‘Akizuki’ pears.

According to a review of the literature, the Raman spectral band at 300–500 nm is the main characteristic peak of cellulose, and the main contributions are the 331 nm, 344 nm, 381 nm, 437 nm, 459 nm, and 520 nm Raman bands at the C-C-C, C-O-C, O-C-C, and O-C-O backbone bending vibrations of cellulose; the C-C-H and O-C-H bending vibrations; and the C-C and C-O skeleton stretching vibrations, mainly the pyran ring outer and bending vibrations. The characteristic peak of lignin in Raman spectroscopy is at approximately 1600 nm, which is attributed primarily to the symmetric stretching vibration of the aromatic ring at 1597 nm, the C=C stretching vibration at 1621 nm, and the C=O stretching vibration at 1662 nm [38,39,40].

In the experiment, Raman spectra were collected in the range of 0–2500 nm. The original spectral images of diseased, asymptomatic, and healthy fruits are shown in Figure 4a, while the average Raman spectra of the three types of fruits are presented in Figure 4b. As illustrated in Figure 4b, the spectral curves of diseased, asymptomatic, and healthy fruits exhibit considerable overlap within the wavelength ranges of 950–1500 nm and 1800–2500 nm. However, significant distinctions are observed in the spectral profiles between diseased and both asymptomatic and healthy fruits within the range of 300–650 nm; notably, no substantial differences exist between asymptomatic and healthy fruits. Furthermore, pronounced variations in the spectral curves among diseased, asymptomatic, and healthy fruits are evident within the range of 1600–1700 nm. Therefore, the changes in the Raman characteristic peaks of these samples are generally consistent with conclusions drawn in other studies.

Figure 4b also shows that the trends of the three spectral curves are becoming consistent, but the reflectances in each band are generally in the order of disease > asymptomatic > healthy. The disease curve shows a significant difference in the Raman characteristic peak range of cellulose and lignin compared with the healthy curve, whereas the asymptomatic samples, although not significantly different, still present some distinctions. This is because asymptomatic fruits are less severely affected than diseased fruits. However, even if the surface of the pear fruit shows no symptoms, the internal damage from cork disease still leads to an increase in lignin and cellulose, which differs from healthy pear fruits. These findings verify the conclusions of Cui et al. [6] that the lignin and cellulose contents in the fruit affected by ‘Akizuki’ pear corky disease are greater than those in healthy fruit. This also provides a basis for the feasibility of analyzing through Raman spectroscopy and establishing accurate classification models.

An essential component of spectral analysis is spectral preprocessing. We can acquire more precise and trustworthy spectrum data via this approach, which offers strong support for further examinations and uses. Different preprocessing techniques affect spectral pictures differently. Figure 5 shows Raman images from nine preprocessing methods. The spectral wavelength (nm) and absorbance (AU) were used as the *x*- and *y*-axes, respectively.

Figure 5 shows that the MSC corrects the scattering effect in the spectral image to make the spectral data closer to the real chemical information. In the processed image, the spectral lines are more consistent, and the scattering-related spectral differences are corrected. The SNV makes the spectral image more homogeneous by standardizing each spectrum and eliminating multiplicative noise and intensity variations. In the processed image, the intensity differences in the spectra are eliminated, and the spectral lines are normalized to the same scale for easy comparison and further analysis. FD highlights changes in the position of absorption or emission peaks in a spectral image, emphasizing the slope of the spectral curve. The processed image shows more edge information, and the position of the peaks becomes more pronounced, but this may result in a weakening of the relative change in peak intensity. SD converges with the function of FD, but its processed image amplifies the noise and loses more information about the peak intensities. SG filtering and WT are mainly used to remove high-frequency noise from spectral images to make the spectral curve smoother. The processed image has less noise and a smoother spectral curve, but it slightly changes the position and shape of the peaks.

The spectral curves obtained by preprocessing, although each has its own characteristics, cannot be compared more intuitively in terms of their effects on the sample data. Therefore, in this study, principal component analysis (PCA) was used to process the results of different preprocessing methods. Reducing the dimensionality of the data while maintaining the most crucial information in the dataset is the primary objective of PCA [41]. The benefits include simplifying the dataset, removing noise and redundancy, and increasing the efficiency and effectiveness of subsequent processing steps. PCA projects the original data onto a new set of axes through a linear transformation. These axes represent the main elements of the information and are arranged in order of variance from largest to smallest. In PCA, data are visualized by selecting the data’s first two main components that contain more information than the two axes, horizontal and vertical. Figure 6 displays the sample distribution under various pretreatments, with H representing a healthy sample, K representing an asymptomatic sample, and Z representing a diseased sample.

Figure 6 illustrates that the sample points of the healthy, asymptomatic, and diseased samples show a certain degree of differentiation between the different categories of samples in the PCA space, although there is a majority degree of overlap. However, the two-dimensional plane’s point distribution of the 345 samples significantly changed following the application of several preprocessing procedures to the original spectral data. A comparative analysis of the two-dimensional distribution maps of 24 samples revealed that WT, SG, and RS presented a high degree of similarity, which was also proven by their corresponding spectral maps. Similarly, the effects of SNV and SG+SNV, MSC and SG+MSC, and FD and SD on the Raman spectra also significantly differ.

### 3.2. Model Optimal Preprocessing Analysis and Comparison

In this study, five classification algorithms under nine Raman spectrum preprocessing methods are compared to find the best algorithm. The accuracies of SVM, RF, GBDT, XGBoost, and CNN on the test set under nine different preprocessing methods are displayed in Figure 7.

The comparative analysis in Figure 7 clearly reveals the key role of the preprocessing method in improving the accuracy of the models. The five models without preprocessing have similar performance and low overall accuracy on the original data, especially SVM and CNN, which instead decrease in accuracy after preprocessing. This is most likely attributed to the improper parameter configuration, resulting in the key features in the original data being inappropriately filtered or lost during preprocessing, which in turn reduces the match with the model. In contrast, the GBDT and RF models showed a significant increase in accuracy after SG+WT preprocessing, indicating that appropriate preprocessing methods can effectively mitigate the noise and undesirable factors in the raw data, thus enhancing the predictive ability of the models.

Specifically, for the choice of preprocessing methods, although SNV can eliminate the magnitude differences between spectra and enhance data comparability, its normalization process may be accompanied by partial loss of spectral information. SG filtering, on the other hand, smooths the data by means of local polynomial fitting, which reduces the noise while keeping the shape and features of the spectral signals better, showing the advantage of retaining the spectral information. On the other hand, WT is known for its excellent noise identification and removal ability, which can retain useful spectral information while eliminating noise that is not useful for classification or analysis.

The combined use of SG and WT becomes an efficient preprocessing strategy for spectral data. First, SG filtering performs preliminary smoothing of the spectral data to initially reduce the interference of noise; subsequently, the WT further refines the processing to identify and remove the remaining noise while ensuring that the key features in the spectral data are retained. This combination of methods not only effectively suppresses noise but also maximizes the retention of detailed information in the spectral data, which is particularly important for Raman spectral analysis that relies on high-resolution spectral features. Therefore, choosing an appropriate combination of preprocessing methods is crucial for improving the accuracy and reliability of the model in spectral data analysis.

Distinct preprocessing methods have different effects on different model performances, and evaluating model preprocessing methods only by a single evaluation index is not comprehensive enough. Thus, using the relative optimal index formula (2) as a basis, this study computed the accuracy and kappa values of each model under various pretreatment methods, and the results were derived by adding the descending ranking of their respective values. In the end, the best pretreatment method for this experiment was determined to be the one with the lowest optimum index. In Table 4, RA denotes the accuracy ranking, and RK denotes the kappa-based ranking of the classification consistency level.

The ROIndex of each preprocessing method is summed by the values in Table 4. The ROIndex values of the nine pretreatment methods are displayed in Figure 8.

Figure 8 illustrates the preprocessing methods on the horizontal axis and their relative optimality indices on the vertical axis. This bar chart clearly shows the performance of different preprocessing methods on the relative optimality index. The figure shows that the FD and SD methods yield the highest values, indicating that these methods strongly influence the original spectral information and are the worst-performing preprocessing methods. In contrast, the SG and SG+WT methods demonstrate the most favorable performance, with relative optimality indices lower than those of RS (original data). This analysis serves as a crucial point of reference for selecting suitable preprocessing methods for data preparation. Moreover, the performance of other preprocessing methods also provides reference information, which helps to comprehensively evaluate the effectiveness of different preprocessing methods.

Among all the preprocessing methods, SG and SG+WT are the best preprocessing methods. As shown in Table 5, when the SG and SG+WT preprocessing methods were used instead of RS, the model performance changed greatly. The accuracy and kappa coefficient of the RF, GBDT, and XGBoost models were greatly enhanced, whereas those of the SVM and CNN models slightly decreased.

### 3.3. Optimization of the Optimal Preprocessing Method for the ‘Akizuki’ Pear Model

In SVM model tuning, the primary parameters that require adjustment include the following key terms: kernel (kernel function), C (penalty coefficient), and gamma (kernel function coefficients). The kernel is used to map the input data into a high-dimensional space, thereby determining the best hyperplane within that space for classification purposes. The linear kernel function (Linear), polynomial kernel function (Poly) and radial basis function Kernel (RBF) are examples of frequently used kernel functions. The choice of a suitable kernel function is crucial for achieving optimal performance in classification problems, as different kernel functions are better suited to different data distributions. The penalty coefficient, C, balances the weights assigned to the classification interval margin and misclassified samples in the objective function. A larger C-value indicates that the model is more focused on fitting each sample, which may lead to overfitting. Conversely, a smaller C-value indicates that the model is more focused on interval boundaries, which can lead to underfitting. Gamma determines the shape and range of the kernel function. The larger the value of gamma is, the smaller the effect of individual samples on the whole classification hyperplane; conversely, the smaller the value of gamma is, the more complex the model.

All the integrated models select “max_depth” and “n_estimators” as the debugging objects. Insufficient n_estimators can result in underfitting, whereas excessive n_estimators may increase the number of computations and are prone to overfitting. The max_depth parameter restricts the depth of the tree, preventing overfitting and improving the model’s generalization ability. It also controls the complexity of the tree, allowing for the capture of more complex patterns. However, a deeper tree may also lead to overfitting. GBDT will also choose the “subsample”, which increases the diversity of the model by randomly selecting some samples instead of all to train each tree, thus improving the model’s performance. In addition, “min_samples_split” limits the conditions under which nodes continue to be partitioned, and “min_samples_leaf” limits the minimum number of samples required for leaf nodes. Higher values of both parameters prevent overfitting but may result in underfitting. To prevent overfitting, XGBoost selects the minimum sum of child node weights, which is referred to as “min_child_weight”.

In the process of CNN model tuning, the main parameters that need to be adjusted are the “batch size” and “epochs”. The number of samples used in each update of the model parameters is determined by the batch size, which needs to be selected according to the specific hardware conditions and the size of the dataset to choose the appropriate batch size. The number of times the model runs through the training set is indicated by the number of epochs. The number of iterations can be increased to improve the model’s performance. However, overfitting can occur when too many iterations are used.

Table 6 lists the precision and kappa coefficient of various models following parameter optimization under the optimal preprocessing methodology. As shown in Table 6, the preprocessing and parameter debugging procedures resulted in enhanced accuracy for the various models. After preprocessing and parameter debugging, SVM, CNN, RF, AdaBoost, GBDT, and XGBoost yield accuracies of approximately 0.74. Among these models, XGBoost following SG+WT processing and RF following SG processing are identified as the best in the context of this experimental model.

To provide a clearer demonstration of the model’s prediction performance for various sample categories, we utilize the confusion matrix to better illustrate the XGBoost and RF classifications. Figure 9 displays the confusion matrix of the two models.

As shown in Figure 9, XGBoost and the RF model predicted a total of 104 samples. Among them, the RF model successfully predicted 79 samples and misclassified 6 healthy samples, 5 asymptomatic samples, and 14 diseased samples. XGBoost successfully predicted 83 samples and misclassified 4 healthy samples, 6 asymptomatic samples, and 11 diseased samples. Although there were some discrepancies in the predictive accuracy of the RF and SG+WT models when different categories of samples were classified, both models demonstrated satisfactory performance.

For the detection of corky disease in the ‘Akizuki’ pear, our team members used different methods to establish a corky disease identification model. In Zhang Hanhan’s study [32], a micro near-infrared spectrometer was employed, and multiple preprocessing methods were used to process the spectral data. When only the NIR spectra were used for modeling, most of the accuracies were approximately 60–70%, and the model performance was poor. Following preprocessing, each model’s performance was significantly enhanced. The Autokeras model, which was preprocessed via SG, exhibited the highest accuracy of 90%. In this study, a Raman spectrometer was used to collect the spectral data, followed by preprocessing and model reconditioning. Finally, the highest accuracy of 80% is achieved through SG+WT preprocessing for XGBoost modeling.

However, the accuracy of this model is still significantly lower than that of Zhang Hanhan’s research, and the accuracy is not high. As shown in Figure 9, the model demonstrates a higher error probability in predicting healthy samples than in predicting asymptomatic and diseased samples. This discrepancy may be attributed to the inadequate number of healthy samples utilized and the similarity of the surface symptoms between the asymptomatic and healthy samples. Therefore, if the model is improved further, expanding the sample size of the training set to increase the accuracy of the model would be beneficial.

### 3.4. Effects of Model Parameters and Preprocessing Methods on Model Performance

As shown in Figure 10, the accuracy of the SVM and CNN models decreases after preprocessing compared with the original data. However, after model tuning, the level of classification consistency is slightly improved, but the classification accuracy is significantly improved. However, preprocessing has a positive effect on the performance of the RF, GBDT and XGBoost models; their classification accuracy and classification consistency level are significantly improved, classification consistency is improved by one to two orders of magnitude, and accuracy is improved from 0.63 (RF), 0.58 (GBDT) and 0.63 (XGBoost) to between 0.68 and 0.80. However, after parameter tuning, their performance improvement is very weak compared with that of the SVM and CNN. 

Integrated learning models, SVMs, and CNNs exhibit significant differences in model performance and sensitivity to preprocessing methods, which is largely attributed to their unique workings and model structures. Integrated learning models improve the overall model accuracy and stability by combining the prediction results from multiple base learners. Its performance is highly dependent on the diversity and quality of the dataset, as the base learner needs to be trained and predicted on different subsets of data. Effective data preprocessing can significantly improve the distribution and feature relevance of the data, thus enhancing the generalization ability of the integrated learning model. However, if the preprocessing step is not performed properly, it may destroy the original information of the data or introduce noise, which in turn affects the performance of the model.

The SVM is a classifier that is based on interval maximization. However, in most cases, data points belonging to different categories are not clearly separated (it is also clear through Figure 5 of this paper that there is no significant difference in the distributions of healthy, asymptomatic, and diseased samples under different preprocessing methods). The SVM algorithm solves the problem of linear indivisibility in low-dimensional space by mapping the data to a high-dimensional space through the kernel function [42]. Therefore, the performance of SVM largely depends on its parameter settings, and by optimizing these parameters (e.g., regularization parameter C and kernel function parameter), SVM can be used to achieve better classification results on the training set, thus improving the classification accuracy of the model.

A CNN, on the other hand, is a deep learning model that automatically extracts high-level features from data through multilayer convolution and pooling operations. The performance of a CNN is affected by internal parameters such as the network architecture, optimizer selection, and learning rate [43]. In the field of deep learning, especially when the amount of training data is sufficient, excessive preprocessing may introduce the risk of overfitting or underfitting, as preprocessing may alter the original distribution of the data or eliminate subtle features useful for model classification. Lee et al. [44] classified Raman spectra obtained from extracellular vesicles (EVs), compared CNN modeling with raw data with baseline-corrected preprocessed data, and reported that the performance of the preprocessed model degraded, with the accuracy decreasing from 95.22% to 90.89%. The study attributes this to the fact that SNR small spectral features may be eliminated by background correction, which is retained in the raw data, and that this subtle information allows the CNN model to understand more details of the input signal, resulting in higher classification accuracy. Many of the preprocessing steps have a degree of arbitrariness, and the preprocessing may alter the original distribution or characteristics of the data, making it difficult for the model to capture valid information. Some preprocessing steps may destroy correlations or structural information between data, resulting in degraded model performance. The parameters or structure of the model may need to be adjusted to the preprocessed data but fail to do so in a timely or correct manner. These factors may also contribute to the degradation of model accuracy after preprocessing of SVM performance.

Therefore, when selecting preprocessing methods, trade-offs need to be made on the basis of the characteristics of the data and the needs of the model. For integrated learning models, preprocessing methods that can enhance the diversity and accuracy of the dataset should be prioritized; for SVMs, their parameters need to be carefully adjusted, and appropriate preprocessing steps should be considered to improve the data distribution; for deep learning models such as CNNs, preprocessing steps need to be carefully selected to avoid introducing the risk of overfitting or underfitting.

Overall, finding the optimal preprocessing methods and parameters is an iterative and comparative process, which may need to be combined with evaluation methods such as cross-validation to optimize model performance. Moreover, owing to the complexity of the spectral data acquisition environment and the weakness of the Raman effect, different experimental conditions and methods may lead to differences in the results; thus, the preprocessing methods and model parameters need to be flexibly adjusted according to the specific situation in practical applications.

## 4. Conclusions

This study successfully constructed a classification model for cork disease in ‘Akizuki’ pears using Raman spectroscopy data and explored the impact of preprocessing and model parameter selection on model performance. Through preprocessing and parameter tuning, the XGBoost model performed the best, achieving an accuracy, F1 score, and kappa coefficient of 0.80, 1.00, and 0.68, respectively, validating the effectiveness of Raman spectroscopy in detecting corky diseases. Among them, SG and SG+WT are the optimal preprocessing methods in this study, significantly enhancing model performance. The RF, GBDT, and XGBoost models are particularly sensitive to preprocessing, with accuracy improvements exceeding 10% through preprocessing. In contrast, the SVM and CNN models rely more on the selection of internal parameters, with accuracy improving by approximately 15% after parameter adjustments. Therefore, in practical applications, choosing appropriate optimization strategies based on the characteristics of the models is necessary.

These results indicate that the use of Raman technology for nondestructive testing of ‘Akizuki’ pear corky disease is feasible, providing a new approach for nondestructive testing. The focus of future work will be on further optimizing preprocessing and modeling methods to increase the accuracy and stability of the models, thereby providing strong support for the quality improvement and market competitiveness of the ‘Akizuki’ pear industry.

## Figures and Tables

**Figure 1 sensors-24-06324-f001:**
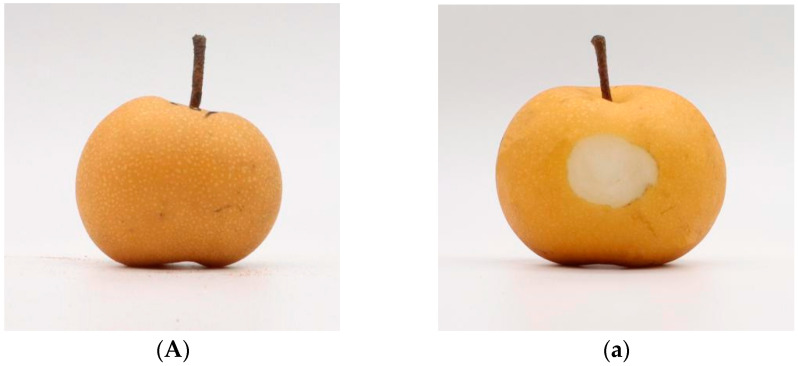
Internal and external comparisons of healthy, diseased, and asymptomatic samples: (**A**) healthy sample surface, (**a**) inside; (**B**) diseased sample surface, (**b**) inside; (**C**) asymptomatic sample surface (**c**) inside.

**Figure 2 sensors-24-06324-f002:**
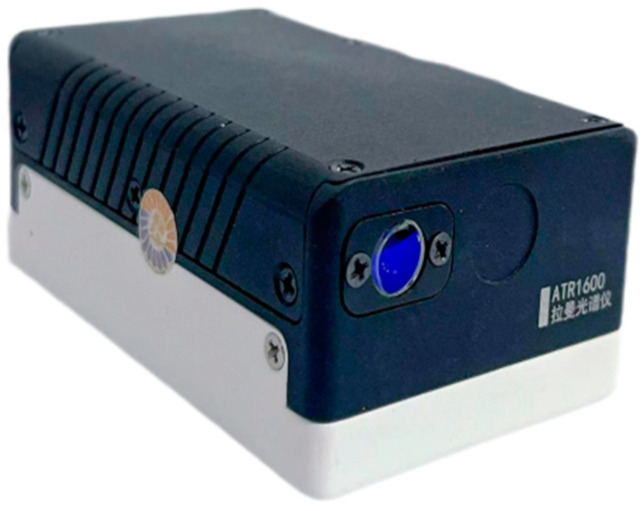
Ultra Miniature Raman Spectrometer “ATR1600” by OPTOTECNICA.

**Figure 3 sensors-24-06324-f003:**
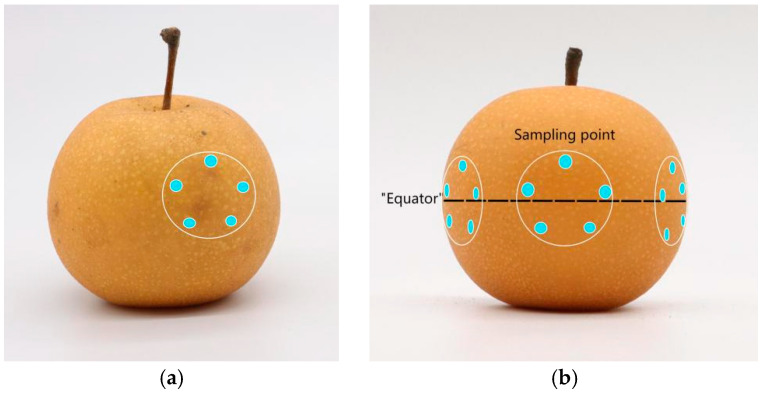
Five-point sampling method for pear fruit: (**a**) sampling method for diseased fruit; (**b**) sampling method for disease-free spot fruit on the surface.

**Figure 4 sensors-24-06324-f004:**
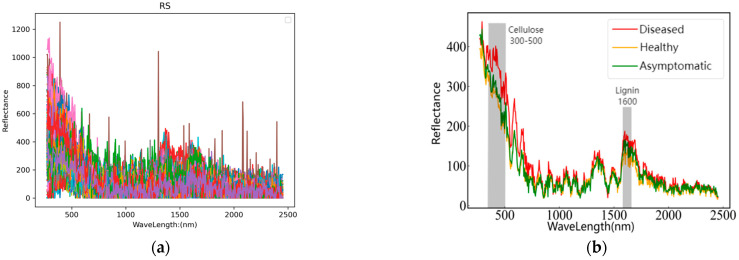
(**a**) Original spectrogram; (**b**) Aaverage spectrogram.

**Figure 5 sensors-24-06324-f005:**
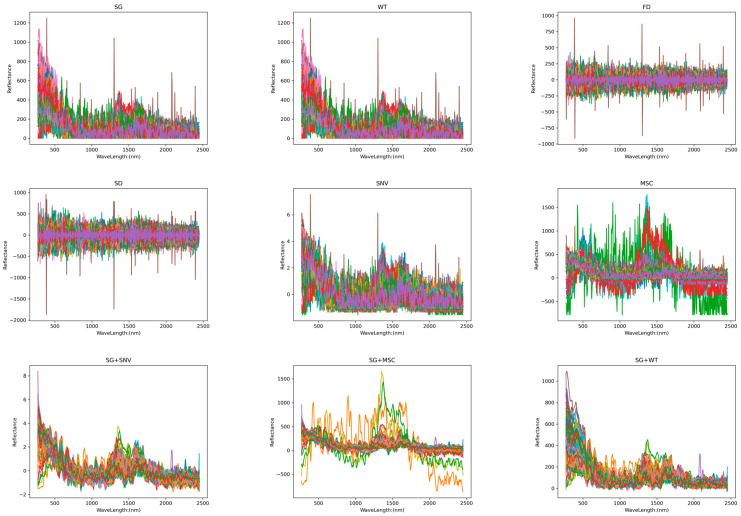
Raman spectra after different pretreatment transformations.

**Figure 6 sensors-24-06324-f006:**
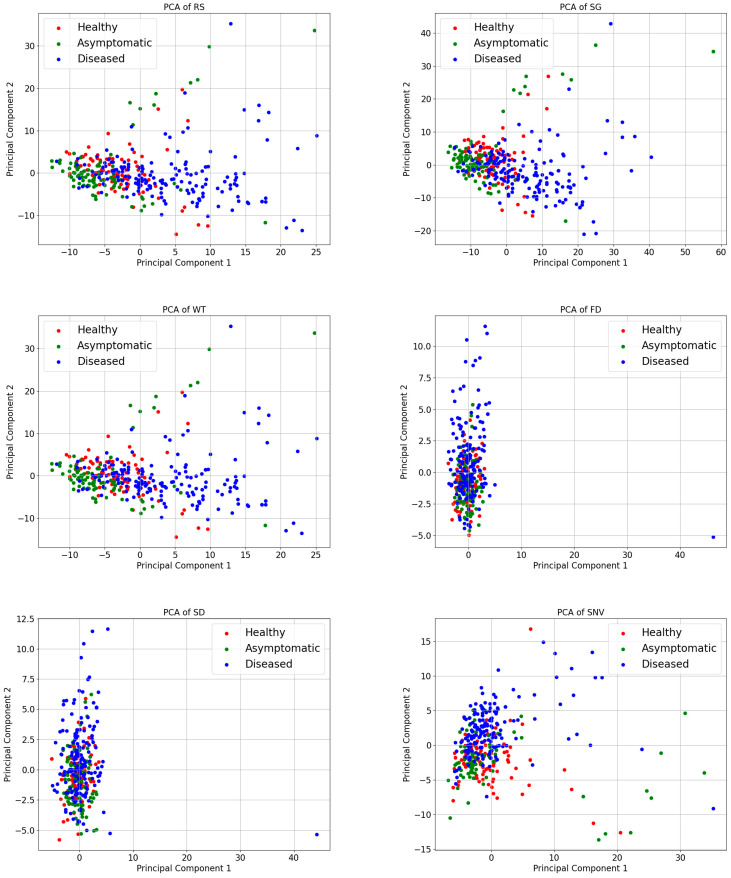
PCA visualization of spectra under different pretreatment methods.

**Figure 7 sensors-24-06324-f007:**
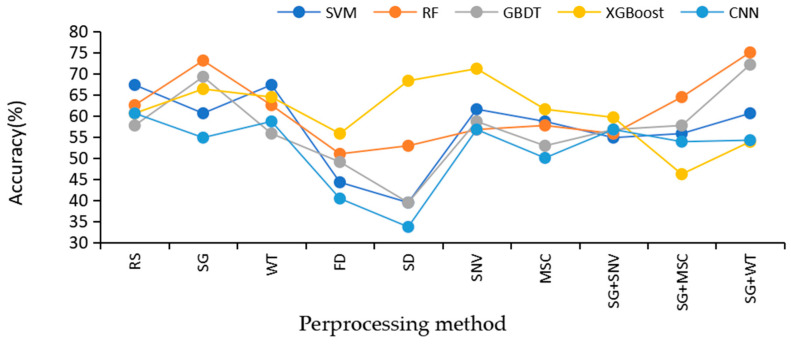
The model was accurate under nine different pretreatment methods.

**Figure 8 sensors-24-06324-f008:**
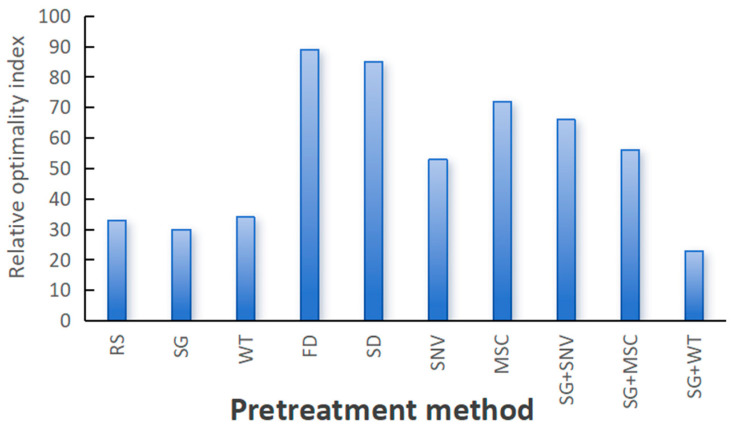
The relative optimization index of the preprocessing method.

**Figure 9 sensors-24-06324-f009:**
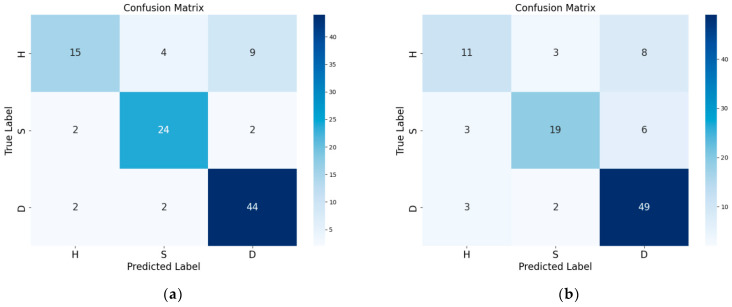
Confusion matrix of the optimal model: (**a**) RF using the pretreatment technique of SG; (**b**) XGBoost using the pretreatment technique of SG+WT.

**Figure 10 sensors-24-06324-f010:**
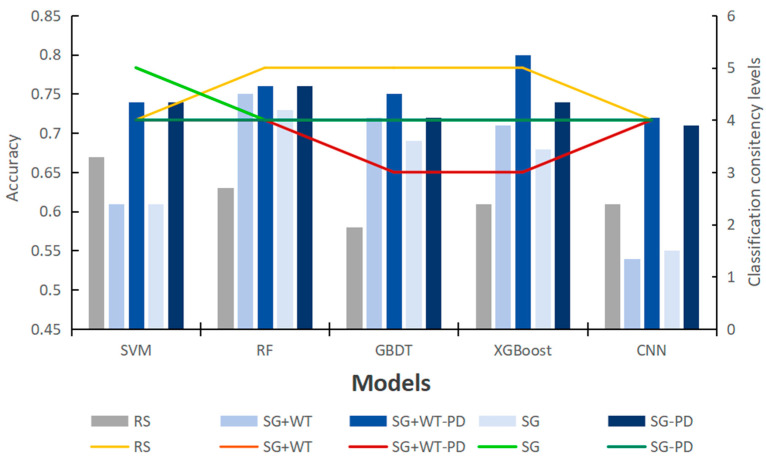
Comparison of model accuracy and classification consistency before and after parameter adjustment.

**Table 1 sensors-24-06324-t001:** Statistics for the ‘Akizuki’ pear sample.

Type	Total Number	Healthy	Asymptomatic	Diseased
Total number of samples	345	87	90	168
Training set	241	65	62	114
Test set	104	22	28	54

**Table 2 sensors-24-06324-t002:** Confusion matrix for the binary classification problem.

Class Designation		Actual Class	
		True (1)	False (0)
Predicted class	Positive (1)	TP	FP
	Negative (0)	TN	FN

**Table 3 sensors-24-06324-t003:** Consistency levels of classifications that match the kappa coefficient.

Kappa Value Range	Classification Consistency	Classification Consistency Levels
<0	Totally Inconsistent	7
0.0–0.20	Slight	6
0.21–0.39	Fair	5
0.40–0.59	Weak	4
0.60–0.79	Moderate	3
0.80–0.90	Substantial	2
0.9	Almost Perfect	1

**Table 4 sensors-24-06324-t004:** The accuracy and kappa coefficient of each model under different preprocessing methods.

Pretreatment Method	SVM		RF		GBDT		XGBoost		CNN	
	RA	RK	RA	RK	RA	RK	RA	RK	RA	RK
RAW	2	2	4	5	5	3	6	4	1	1
SG	5	5	2	2	2	1	2	2	5	4
WT	1	1	5	3	7	6	3	3	2	3
FD	9	9	10	10	9	9	10	9	9	5
SD	10	8	9	6	10	5	9	10	10	8
SNV	3	3	7	8	3	7	4	6	3	9
MSC	6	6	7	7	8	8	7	8	8	7
SG+SNV	8	8	8	6	6	5	8	7	4	6
SG+MSC	7	7	3	4	4	4	5	5	7	10
SG+WT	4	4	1	1	1	2	1	1	6	2

**Table 5 sensors-24-06324-t005:** Model accuracy and kappa coefficient under the optimal preprocessing method.

Pretreatment Method	RS		SG+WT		SG	
	Accuracy	Kappa	Accuracy	Kappa	Accuracy	Kappa
SWM	0.67	0.48	0.61	0.38	0.61	0.38
RF	0.63	0.32	0.75	0.58	0.73	0.51
GBDT	0.58	0.34	0.72	0.51	0.69	0.52
XGBoost	0.61	0.35	0.71	0.59	0.68	0.51
CNN	0.61	0.52	0.54	0.49	0.55	0.46

**Table 6 sensors-24-06324-t006:** Model parameter values, accuracy of the optimized model and kappa values.

Pretreatment Method	SG+WT				SG			
	Parameters	Accuracy	F1	Kappa	Parameters	Accuracy	F1	Kappa
SVM	Kernel = poly	0.74	0.69	0.57	Kernel = poly	0.74	0.69	0.57
	C = 1				C = 1			
	gamma = scale				gamma = scale			
RF	max_depth = 11	0.76	0.71	0.59	max_depth = 11	0.76	0.71	0.59
	n_estimators = 68				n_estimators = 49			
GBDT	max_depth = 5	0.75	1.00	0.60	max_depth = 3	0.72	1.00	0.55
	n_estimators = 150				n_estimators = 50			
	Subsample = 0.6				Subsample = 0.6			
	min_samples_leaf = 4				min_samples_leaf = 2			
	min_samples_split = 10				min_samples_split = 2			
XGBoost	max_depth = 7	0.80	1.00	0.68	max_depth = 3	0.74	1.00	0.58
	n_estimators = 200				n_estimators = 50			
	Subsample = 0.8				Subsample = 0.6			
	min_child_weight = 1				min_child_weight = 5			
CNN	Batch_size = 64	0.72	0.71	0.56	Batch_size = 64	0.71	0.70	0.54
	epochs = 11				epochs = 11			

## Data Availability

The original contributions presented in the study are included in the article, further inquiries can be directed to the corresponding author/s.

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
