# Peer review of "Nondestructive Detection of Corky Disease in Symptomless ‘Akizuki’ Pears via Raman Spectroscopy"

_sensors, 2024, doi:10.3390/s24196324_

Round 1

Reviewer 1 Report

Comments and Suggestions for Authors

This manuscript used the Raman spectroscopy to o develop an early diagnosis model of Akizuki' pear (Pyrus pyrifolia Nakai) corky disease.. The results show that RF, GBDT, and XGBoost are more sensitive to the pretreatment method, whereas SVM and CNN are more dependent on internal parameter tuning. The results of this study indicate that the early detection of Raman spectroscopy represents a novel approach for the nondestructive identification of asymptomatic 'Akizuki' pear corky disease.

This manuscript has the good novelty. But there were a few question:

1. Why is it called a asymptomatic? The samples should be sliced to show internal state of pear.

2. In the figure 6, the legend is completely unreadable. You should add more explanation for this PCA figure.

3. In the figure 7, the Y-axis had no units.

4. The table 6 was not necessary, therefore it could be deleted.

5. The words in the figure 9 were too small to read clearly.

The content and the structure of the manuscript were clear. But there is no detailed explanation on the physicochemical composition changing of the disease, especially how the Raman spectrum could be associated with the physicochemical composition of pear, therefore the result should add this content. The conclusion should be a summary of your results, not simply a list of items. It needs further optimization and improvement.

Reviewer 2 Report

Comments and Suggestions for Authors

The article “Nondestructive detection of corky disease in symptomless 'Akizuki' pears via Raman spectroscopy” reported a Raman spectroscopy-based corky disease recognition which is enhanced with various data process methods. However, there are some essential issues in the submitted articles. It is my opinion that the current manuscript is not recommended to Sensors. Detailed comments are:

  1. The reliability of the data process-based identification depends on the quality of the data. However, the authors collected data from fruits “selected under the guidance of experienced local fruit farmers.”. Considering that the authors also stated that “The disease cannot be identified through visual inspection alone, …. ”, it is wondered beyond the guidance whether there are any other persuading identifications to confirm the selection of healthy, diseased and asymptomatic samples before the Raman tests. The confirmation could guarantee the Raman data from the corresponding fruit at three different statuses. If the identification of the fruits before Raman measurement is not solid enough, the following work would be questioned.
  2. It is questioning how the sampling points were determined. It seems in Figure 3 the Raman spectra were collected from the points around the brown spots of diseased fruits. Meanwhile, in the article, the statement “The disease cannot be identified through visual inspection alone” suggests that the brown spots are not always the diseased location. Then how did the authors confirm that the brown spots they chose for Raman tests are the diseased location?
  3. It is confusing with the statement “Asymptomatic fruit will have brown concave round spots on the surface, which can be observed by the naked eye.” Since there are “brown concave round spots”, it is not “asymptomatic”.
  4. An interesting fact was noted that according to Figure 1, the “experienced local fruit farmers” seem to be able to identify healthy, diseased and asymptomatic fruits in a non-destructive and on-site way? In contrast, the reported method requires fruit harvesting, pretreatment, Raman test and data processing.
  5. Some details should be updated or checked, for instance:

  1. More details on the Raman spectra collection. Integration time and power.
  2. What is the meaning of the sentence “120°C intervals near the equator line of healthy and internally diseased fruits”?
  3. The first picture in Figure 6 labels “0”, “1”, “2” while other label “H”, “K”, “Z”.
  4. It is stated that “As shown in Figure 9, XGBoost and the RF model predicted a total of 104 samples.”. However, there are 90 test samples (90 fruits) as stated in Section 2.1. 

Reviewer 3 Report

Comments and Suggestions for Authors

The study utilizes Raman spectroscopy coupled with machine learning models, including SVM, RF, GBDT, XGBoost, and CNN, to diagnose 'Akizuki' pear corky disease early. The authors stated that the optimal preprocessing with SG+WT enhanced model performance, with XGBoost achieving 80% accuracy. However, the current manuscript lacks novelty and a comprehensive literature review. Some concerns must be addressed to meet the publication standard.

  1. From the introduction, the main contribution or novelty of this study is unclear. Additionally, the combination of Raman spectroscopy and machine learning has been extensively explored in numerous research papers. A more detailed and comprehensive literature review is needed to contextualize this study. The following papers and more should be discussed. 10.1016/j.talanta.2021.122195, 10.1016/j.measurement.2023.113121, 10.1016/j.foodchem.2021.131471.
  2. What is the point of including Figure 1 in the introduction? It is not clear to have such a figure without explanation in the introduction section.
  3. In section 2, what criteria were used for selecting the sampling areas on the fruits, and how does this ensure that the data are representative of the diseased and asymptomatic conditions?
  4. What are the specific features in the Raman spectra that lead to improved classification accuracy after certain preprocessing methods (e.g., SG+WT) compared to others (e.g., SNV or MSC)?
  5. Why do some models (e.g., SVM, CNN) show reduced accuracy after preprocessing, and how could this issue be addressed in future work?
  6. Why were the accuracy improvements in SVM and CNN minimal despite parameter tuning, while integrated models showed more significant enhancements?
  7. Figures 4, 5, 6, and 9 are blur. Improve the resolution and enlarge the font size for better visualization. And Figure 5 misses legends. 

Comments on the Quality of English Language

In general, the current version contains some typos and grammatical errors. All the page numbers at the bottom are wrong. A comprehensive proofreading is necessary to rectify these issues and enhance the manuscript's clarity and professionalism.

Round 2

Reviewer 2 Report

Comments and Suggestions for Authors

The revised article “Nondestructive detection of corky disease in symptomless 'Akizuki' pears via Raman spectroscopy” has addressed the comments. It is my opinion that the current manuscript could be published in Sensors.

Reviewer 3 Report

Comments and Suggestions for Authors

The authors have addressed my concerns.